# RETRACTED: Characterization of the Anti-Biofilm and Anti-Quorum Sensing Activities of the β-Adrenoreceptor Antagonist Atenolol against Gram-Negative Bacterial Pathogens

**DOI:** 10.3390/ijms232113088

**Published:** 2022-10-28

**Authors:** Simona Cavalu, Samar S. Elbaramawi, Ahmed G. Eissa, Mohamed F. Radwan, Tarek S. Ibrahim, El-Sayed Khafagy, Bruno Silvester Lopes, Mohamed A. M. Ali, Wael A. H. Hegazy, Mahmoud A. Elfaky

**Affiliations:** 1Faculty of Medicine and Pharmacy, University of Oradea, P-ta 1 Decembrie 10, 410087 Oradea, Romania; 2Medicinal Chemistry Department, Faculty of Pharmacy, Zagazig University, Zagazig 44519, Egypt; sselbaramawy@pharmacy.zu.edu.eg (S.S.E.); ageissa@pharmacy.zu.edu.eg (A.G.E.); 3Department of Pharmaceutical Chemistry, Faculty of Pharmacy, King Abdulaziz University, Jeddah 21589, Saudi Arabia; mfarghaly@kau.edu.sa (M.F.R.); tmabrahem@kau.edu.sa (T.S.I.); 4Department of Pharmaceutics, College of Pharmacy, Prince Sattam Bin Abdulaziz University, Al-kharj 11942, Saudi Arabia; e.khafagy@psau.edu.sa; 5Department of Pharmaceutics and Industrial Pharmacy, Faculty of Pharmacy, Suez Canal University, Ismailia 41522, Egypt; 6School of Health and Life Sciences, Teesside University, Middlesbrough TS1 3BA, UK; brunoldlopez@gmail.com; 7National Horizons Centre, Teesside University, Darlington DL1 1HG, UK; 8Department of Biology, College of Science, Imam Mohammad Ibn Saud Islamic University, Riyadh 11432, Saudi Arabia; mamzaid@imamu.edu.sa or mohd_ali2@sci.asu.edu.eg; 9Department of Biochemistry, Faculty of Science, Ain Shams University, Abbassia, Cairo 11566, Egypt; 10Department of Microbiology and Immunology, Faculty of Pharmacy, Zagazig University, Zagazig 44519, Egypt; waelmhegazy@daad-alumni.de; 11Pharmacy Program, Department of Pharmaceutical Sciences, Oman College of Health Sciences, Muscat 113, Oman; 12Department of Natural Products, Faculty of Pharmacy, King Abdulaziz University, Jeddah 21589, Saudi Arabia; 13Centre for Artificial Intelligence in Precision Medicines, King Abdulaziz University, Jeddah 21589, Saudi Arabia

**Keywords:** atenolol, Quorum sensing, anti-biofilm, virulence, Gram-negative, antimicrobial resistance

## Abstract

The development of bacterial resistance to antibiotics is an increasing public health issue that worsens with the formation of biofilms. Quorum sensing (QS) orchestrates the bacterial virulence and controls the formation of biofilm. Targeting bacterial virulence is promising approach to overcome the resistance increment to antibiotics. In a previous detailed in silico study, the anti-QS activities of twenty-two β-adrenoreceptor blockers were screened supposing atenolol as a promising candidate. The current study aims to evaluate the anti-QS, anti-biofilm and anti-virulence activities of the β-adrenoreceptor blocker atenolol against Gram-negative bacteria *Serratia marcescens*, *Pseudomonas aeruginosa*, and *Proteus mirabilis*. An in silico study was conducted to evaluate the binding affinity of atenolol to *S. marcescens* SmaR QS receptor, *P. aeruginosa* QscR QS receptor, and *P. mirabilis* MrpH adhesin. The atenolol anti-virulence activity was evaluated against the tested strains in vitro and in vivo. The present finding shows considerable ability of atenolol to compete with QS proteins and significantly downregulated the expression of QS- and virulence-encoding genes. Atenolol showed significant reduction in the tested bacterial biofilm formation, virulence enzyme production, and motility. Furthermore, atenolol significantly diminished the bacterial capacity for killing and protected mice. In conclusion, atenolol has potential anti-QS and anti-virulence activities against *S. marcescens*, *P. aeruginosa*, and *P. mirabilis* and can be used as an adjuvant in treatment of aggressive bacterial infections.

## 1. Introduction

Quorum sensing (QS) is the chemical language that bacteria use to communicate with each other in populations to arrange their accommodation and invasion in host tissues [1,2,3,4]. It relies on diverse chemical signaling molecules called autoinducers (AIs), which are acyl-homoserine lactones (AHL) in Gram-negative pathogens or peptides in Gram-positive [1,5,6,7]. However, AHL can passively diffuse through the Gram-negative thin cell wall, the Gram-positive peptide autoinducers must be actively transported [2,5]. Once, autoinducers bind to their cognate receptors, the formed receptor-AI regulate the expression of genes involved in virulence, biofilm formation, conjugation, sporulation, bioluminescence, and competence [1]. Like languages between humans, these signals differ between species; some bacteria can interpret several signals, while others sense a select few [1,2,5,8,9]. For instance, most Enterobacteriaceae utilize mainly Lux-type QS receptors to sense a wide diversity of AIs [9,10,11,12]. Furthermore, there is QS communication between different bacterial species, some species cannot produce their own AIs, but have receptors for the AIs of others [1,2,13]. Interestingly, interference with the QS systems by hindering the involved QS receptors lead to significant diminishing of bacterial virulence [14,15,16]. Because of its crucial role in controlling the bacterial virulence, targeting QS was supposed as a promising approach to attenuate the bacterial pathogenesis to be easily eradicated without developing bacterial resistance [3,17]. 

Bacterial resistance constitutes a serious global health issue, as bacteria magnificently developed biochemical and genetic resistance to nearly all antibiotic classes [18,19]. This situation dictates finding out new antibiotics and developing newer approaches to conquer this resistance increment [18,20,21,22,23,24]. The bacterial resistance was observed tremendously among different species of Gram-negative bacteria which can share the resistance genes [20,25,26]. Furthermore, the bacterial ability to form biofilms worsens the situation and increase the resistance to almost all antibiotics [20,27,28]. In this context, alleviation of bacterial virulence was suggested as a promising approach [2,16,29,30]. This approach confers the attenuation of bacterial virulence to be easily cleared without stressing the bacteria to develop resistance since it does not affect bacterial growth [16,30,31]. QS controls bacterial virulence such as biofilm formation, motility, production of enzymes and other virulence factors [32,33,34,35]. Due to the key roles of QS, several studies screened the anti-QS activities of various chemical moieties [36,37,38], natural products [39,40], and approved safe drugs [41,42,43,44]. 

Drug repurposing merits have been approved and it is considered an important strategy to save efforts, time and costs [30,35,45,46]. In a previous study the anti-QS activities of twenty-two β-adrenoreceptor blockers were screened in silico, and they were considered as promising anti-virulence candidates. Furthermore, metoprolol showed anti-virulence activities in vitro and in vivo against *Pseudomonas aeruginosa* and *Salmonella* Typhimurium [42]. Almalki. et al., performed a detailed docking study of the tested twenty-two β-adrenoreceptor blockers on the main structurally different Lux-type QS receptors *Agrobacterium tumefaciens* (TraR; PDB entry: 1L3L), *Pseudomonas aeruginosa* (QscR; PDB entry: 3SZT), and *Chromobacterium violaceum* (CviR; PDB entry: 3QP5) analyzing the binding interactions of the receptor ligands. Furthermore, a molecular dynamics simulation was performed to support the docking study findings, and atenolol was shown as a promising anti-QS candidate [42]. Atenolol as well as metoprolol are used to treat the hypertension and related complications [47]. Furthermore, atenolol shares the metoprolol chemical moiety 1-[4-methylphenoxy]-3-(propan-2-ylamino) propan-2-ol, as shown in Figure 1, that encourages us to investigate the atenolol’s anti-QS and anti-virulence activities. The current study aims to evaluate the anti-QS activities of β-adrenoreceptor blocker atenolol against three of the most common Gram-negative nosocomial pathogens, *Serratia marcescens*, *P. aeruginosa*, and *Proteus mirabilis*. The atenolol binding affinity to different QS protein targets were examined in silico. Further, the anti-virulence activities of atenolol against the three bacteria were assessed in vitro and in vivo. 

## 2. Results

### 2.1. In Silico Molecular Study

#### 2.1.1. Screening of Atenolol Binding on Different QS Biotargets

A two-step docking protocol, consisting of a preliminary rigid receptor approach and a further induced-fit docking, was applied to explore the interactions of atenolol with quorum sensing pathway components. The interactions with three bacterial targets regulating virulence genes of three different bacteria were screened; the *P. aeruginosa* QS signal receptor QScR (PDB ID: 3SZT) and the *P. mirabilis* adhesin MrpH (PDB ID: 6Y4F), as well as a SWISS model for the *S. marcescens* SmaR (Uniprot entry: Q14RS3) characterized the targets for this in silico evaluation. 

The binding pockets of the bio-targets were defined by matching the location of the co-crystallized ligand with the sites obtained from MOE Site Finder. Figure 2A–C shows the three-dimensional protein architecture and the putative binding pockets with the calculated Richards’ solvent accessible surface area (SA) and volume (V) of the three bio-targets as predictable using the Computed Atlas for Surface Topography of Proteins (CASTp).

The *P. aeruginosa* quorum-sensing transcription factor (QscR) was resolved as homodimer at resolution of 2.55 Å with *N*-3′oxo-octanoyl-L-homoserine lactone (3OC12-HSL) co-crystallized within the active site. The architecture of QscR showed the same α/β/α sandwich as *E. coli* SdiA. The Ligand-binding domain (LBD) and the DNA-binding domain (DBD) are also linked together by a 10 amino acid-short sequence of residues [9]. Figure 2D illustrates the main amino acid residues lining the surface of the binding pocket. 

The *P. mirabilis* adhesin MrpH architecture shows seven β-strands and two α helices in the 1.75Å crystal structure. The structure of MrpH is characterised by the close proximity between the N-terminus and the C-terminus, a disulfide bond between Cys 128 and Cys 152, which is an integral part for the protein structure and function, and the presence of a Zn ion bound by His 72, His 74, His 117 and is tetrahedrally coordinated by binding to an external ligand [48,49]. Figure 2E illustrates the main amino acid residues lining the surface of the binding pocket.

A homology model for *S. marcescens* Quorum-sensing transcriptional regulator (SmaR) (Uniprot Entry: Q14RS3, www.uniprot.org (accessed on 28 May 2022)) was built depending upon *Chromobacterium violaceum* CviR transcriptional regulator (PDB code: 3QP5, 3.25 Å) as a template [3], using SWISS model (https://www.expasy.org/resources/swiss-model (accessed on 28 May 2022)) [50,51,52,53,54]. Moreover, CASTP was used to verify the binding pocket. Figure 2F illustrates the main amino acid residues lining the surface of the binding pocket. The active site of the model was identified by the Site Finder module in the MOE which matched with the co-crystallized ligand 4-(4-chlorophenoxy)-N-[(3S)-2-oxooxolan-3-yl]butanamide (HLC) binding site of the template. 

#### 2.1.2. Docking Simulations on *P. aeruginosa* QS Control Repressor

Docking of atenolol and the co-crystallized ligand (3OC12-HSL) with QScR showed comparable fitting in the binding pocket, as shown in Figure 3A. Both of them was able to fill the active site forming hydrophobic interactions with the main amino acids lining the pocket Ser38, Tyr52, Phe54, Tyr58, Tyr66, Val78 and Met127 Figure 3B. However, the dock score of the co-crystallized ligand was much better than atenolol (−10.1774 and −7.5851, respectively)

Atenolol oriented within the active site through H-bond and ionic bind interactions with the crucial amino acid residues Figure 3C. As, the secondary amino group and the hydroxyl group formed H-bond interactions with Gly40, with Tyr85, respectively. In addition, protonated amino group showed ionic bond interaction with acidic Asp75 (Figure 3C). The co-crystallized ligand (3OC12-HSL) showed different H-bond interactions with Tyr58, Trp62 and Asp75. Table 1.

#### 2.1.3. Docking Simulations on *P. mirabilis* Adhesin MrpH

Virtually, atenolol acts as an external ligand for Zn^+2^ binding; crucial for biofilm formation Figure 4. As the amide group allowed atenolol to be oriented in the active site through interaction with zinc metal and formation of hydrogen bond with basic Arg118. Moreover, the phenyl ring interacted with Asn82 through *Pi*-H bond. Protonated amino group formed hydrogen bond with Ala84. This is in addition to the comparable hydrophobic interactions to glutamate (co-crystallized ligand) representing in Asn82, Ala84, Phe85, Thr115 Thr116, Arg118 and Ile140 residues (Figure 4).

Results of the docking process as summarized in Table 2 indicates that glutamate had a relatively lower docking energy score = −8.4332 Kcal/mol than that of atenolol −7.2011 Kcal/mol as glutamate has much smaller size; enabling the molecule to move and bind freely; than a much bigger molecule like atenolol, as shown by the superimposition of both ligands in the MrpH pocket in Figure 5.

#### 2.1.4. Docking Simulations on *S. marcescens* SmaR

After the model was built Figure 6, docking of the ligand co-crystallized with the template was performed to both validate the process and compare to atenolol binding.

Generally, the docking results, summarized in Table 3, showed that the score for atenolol was higher than HLC; however, the various binding interactions resulted in moderate score and stabilization of the complex.

Anionic sidechain of the acidic Asp66 residue participated in polar interaction with the protonated amino group of atenolol as well as H-bond interaction. The atenolol structure showed several *pi*-interactions as the following: H-*pi* bond with Phe44, *pi*-*pi* bond with Tyr57 and cation-*pi* bond with Trp81. Beside these interactions, the ligand had several hydrophobic interactions with Phe44, Phe54, Tyr57, Asp66, Val68, Trp81 and Ile105 residues, participating in ligand stabilization within the pocket Figure 7.

### 2.2. Minimum Inhibitory Concentration (MIC) of Atenolol 

Microtiter broth dilution was used to determine the atenolol MICs, atenolol inhibited the growth of *S. marcescens, P. aeruginosa*, and *P. mirabilis* at concentrations 4, 2, and 2 mg/mL, respectively.

### 2.3. Effect of Atenolol on Bacterial Growth

To avoid the atenolol effect on bacterial growth, the antivirulence activities were tested for atenolol at sub-MIC (1/5 MIC). Furthermore, the atenolol effect on bacterial growth was attested by comparing the optical densities and bacterial counts of control untreated bacterial cultures and atenolol-treated cultures. The experiment was conducted in triplicate there was no significant difference between the bacterial growth in the presence and absence of atenolol at sub-MIC (Figure 8). 

### 2.4. In Vitro Antivirulence and Anti-QS Activities of Atenolol

#### 2.4.1. Antibiofilm Activity

The bacterial adhesion and biofilm formation on inanimate object was quantified with the crystal violet method in the presence or absence of atenolol at sub-MIC. Atenolol significantly decreased the biofilm formation (Figure 9). A representative light microscope images were captured that show a marked decrease in the numbers of adhered biofilm forming bacterial cells (Figure 9A).

#### 2.4.2. Diminishing of Bacterial Motility

The effect of atenolol at sub-MIC on the swarming motility of the tested bacterial strains were assessed. Atenolol significantly decreased the swarming motility in *S. marcescens, P. aeruginosa*, and *P. mirabilis* (Figure 10).

#### 2.4.3. Decreasing Production of Virulence Enzymes

Protease and hemolysins play critical roles in establishing and spreading bacterial infections. The atenolol effect at sub-MIC on the production of extracellular enzymes was evaluated on the production of hemolysins and protease in *S. marcescens, P. aeruginosa*, and *P. mirabilis*. Atenolol significantly decreased the production of hemolysins and protease by tested strains (Figure 11). 

### 2.5. Atenolol Downregulation of Virulence and QS Genes

The expression of *P. aeruginosa* QS encoding genes were evaluated in the treated bacterial samples with atenolol at sub-MIC and compared to untreated controls. Atenolol significantly decreased the expression of the genes encoding autoinducer synthetase and their receptors *lasI/R, rhlI/R,* and *pqsA/R*. Furthermore, atenolol significantly upregulated the expression of the motility controlling system repressor *rsmA* in *S. marcescens* and in same time significantly downregulated the motility controlling genes *rssB,* and flagella encoding genes *flhC, flhD*. Atenolol also downregulated the expression of the fimbria encoding genes *fimA, fimC*, and *bsmB*. These results indicate the significant downregulation of QS encoding, motility, and adhesion genes in parallel to upregulation to the virulence genes repressor RsmA. (Figure 12).

### 2.6. In Vivo Antivirulence and Anti-QS Activities of Atenolol

Mice protection assay was used to evaluate the capability of atenolol at sub-MIC to mitigate the *S. marcescens, P. aeruginosa*, and *P. mirabilis* pathogenesis. Meanwhile, there were no deaths among mice in negative control groups, the death rates were 50%, 80%, and 60% in mice injected with untreated *S. marcescens, P. aeruginosa*, and *P. mirabilis*, respectively. Interestingly, the death rates were significantly decreased in mice groups injected with atenolol treated *S. marcescens, P. aeruginosa*, and *P. mirabilis* to 20%, 40% and 30%, respectively (log rank test for trend *p* = 0.0289, 0.0025, and 0.0111, respectively) (Figure 13).

## 3. Discussion

The impact of the development of bacterial resistance to antibiotics is in increase and constitutes a serious challenge to public health [20,35,55]. The shortage in the discovery of new efficient antibiotics worsens the situation against the renewed bacterial capability to develop resistance [23,35,46]. Despite the necessity of antibiotics, they may be insufficient, especially in aggressive resistant infections [56,57,58]. Adopting new approaches is required to diminish the development of bacterial resistance [19,22,30,59]. Targeting bacterial virulence is one of these approaches that attenuates the bacteria easing their eradication by immunity without stressing bacteria to develop resistance [60,61]. The crucial roles of bacterial QS systems in controlling the bacterial virulence are proven [3,18,62]. Additionally, hence, targeting QS is meaningful to diminish several bacterial virulence factors such as biofilm formation, enzyme production, virulence agents production, and motility [4,6,61,63]. Several chemical compounds and natural products have been screened for their anti-QS activities and anti-virulence activities in vitro and in vivo [17,64]. 

The high attrition rates for the discovery and development of new drugs including costs, and time-consuming make the repurposing of old clinically known drugs an attractive proposition [45,65]. Drug repurposing is the use of already approved safe drugs that guarantee to shorten the development timelines and lower the overall development costs. However, numerous data-driven and experimental approaches have been proposed for repurposing several drugs, there are several challenges that are needed to be overcome as reviewed [45]. One of the efficient approaches to evaluate the possibility of drugs to be repurposed is the detailed in silico investigation. This virtual in silico studies were used efficiently in evaluating the binding abilities of tested drugs to specific protein receptors [41,42,66,67]. In this context, the adrenoreceptor antagonists were docked to several bacterial targets to test their anti-virulence activities [41,42]. Atenolol showed considered binding abilities to structurally different QS receptors that indicate its possible anti-virulence activities [42]. The present study evaluates the anti-QS and anti-virulence activities of atenolol against three Gram-negative bacteria *S. marcescens, P. aeruginosa*, and *P. mirabilis*. 

*P. mirabilis* is peritrichous Gram-negative bacterium known for its characterized swarming motility. *P. mirabilis* is causative agent of diverse serious infections such as burn, wound, urinary tract, diabetic foot, and other infections [7,49,68,69]. *P. mirabilis* ability to gain resistance via horizontal gene transfer explains the increased isolation of resistant isolates in clinics and labs [70,71,72]. *P. mirabilis* acquires considered diversity of virulence factors enabling it to invade and spread into host tissues such as urease, protease and hemolysins besides the motility [49,68,71]. In the current study, a clinical *P. mirabilis* isolate obtained from macerated diabetic foot wound was used [49]. In addition to virulence, the tested *P. mirabilis* was described as strong-biofilm forming and showed resistance to several antibiotic classes [69,72,73]. Furthermore, another clinical *S. marcescens* isolate obtained from endotracheal aspirate was used in this study [74]. The *S. marcescens* clinical importance is owed to its increased share in causing nosocomial infections. *S. marcescens* was documented the seventh and the tenth most common pathogen that causes nosocomial pneumonia and blood stream infections, respectively [75,76,77]. Moreover, there is an exaggerated *S. marcescens* resistance to fluoroquinolones, β-lactam, and aminoglycosides [75,76,78]. Additionally, *P. aeruginosa* was used to evaluate the atenolol anti-virulence activity. *P. aeruginosa* is known for its virulence and resistance to antibiotics and disinfectants [24,79,80]. *P. aeruginosa* arsenal of virulence factors enables it to cause a wide range of serious infections in nearly all tissues [79,81]. Although the three selected Gram-negative bacteria for this study share their clinical importance, they acquire different virulence behavior and utilize different QS systems.

The bacterial virulence is mainly regulated by QS communication system. Bacteria can sense the adrenergic hormones and eavesdrops on the host cells to establish their infection [82,83,84]. Adrenergic hormones were shown to enhance the bacterial virulence [85,86,87,88]. This makes β-adrenoreceptor blockers a promising option to stop bacterial espionage on our cells and inhibit bacterial QS systems. A previous leader study evaluating the affinities of β-blockers towards different QS receptors from three bacteria namely, *Chromobacterium violaceum*, *Pseudomonas aeruginosa* and *Salmonella* Typhimurium, pointed towards the ability of atenolol, esmolol and metoprolol to decrease the production of QS-controlled *C. violaceum* and biofilm formation by *P. aeruginosa* and *S.* Typhimurium along with QS encoding genes downregulation [42]. To further expand on these findings and provide evidence for the β-blockers ability of combating QS in Gram-bacteria, computational and biological evaluation of atenolol against previously studied *P. aeruginosa*, as a linking point of comparison and verification of the here-in findings, and expanding to *S. marcescens* and *P. mirabilis* were performed using three different QS targets from the three bacteria for the in silico studies and various array of biological experimentation to indicate the numerous possibilities and the detrimental effect of atenolol as one of the most promising β-blocker examples against bacterial virulence factors. These findings supported by the wealth of evidence of the benefits of drug repurposing of approved drugs in saving time and cost could indeed revolutionize the way of combating infections and upgrade the available machinery towards Gram-negative infections.

Consequently, a detailed molecular docking study was conducted to evaluate the atenolol ability to compete on essential QS proteins in the tested bacterial strains. Gram-negative QS systems are generally LuxI/R type; *S. marcescens* SmaR is a LuxR family member that controls its virulence such as swarming motility, biofilm production, hemolytic activity, and production of enzymes [74,84,89]. *P. aeruginosa* employs two LuxI/R QS systems namely LasI/R and RhlI/R besides non-Lux type PQS system [81]. In addition, *P. aeruginosa* hires a Lux analogues QscR system to sense *lasI* autoinducers [40,81]. The Mannose Resistant Proteus like fimbriae (MRP) is Zn-dependent receptor-binding domain encoded by operon mrpABCDEFGHJ [72,90]. MRP fimbriae are essential for *P. mirabilis* aggregation and biofilm formation; where MrpH is one of the key involved fimbrial proteins [48,49]. Our findings showed a marked atenolol binding ability to *S. marcescens* SmaR, *P. aeruginosa* QscR, and *P. mirabilis* MrpH that point to possible anti-QS and anti-Virulence activities. It is worthy to mention that our previous study showed atenolol ability to compete on other different Lux-QS receptors *Agrobacterium tumefaciens* TraR and *Chromobacterium violaceum* CviR.

The underlying assumption toward an effective implementation of a successful QS interference is inhibition of virulence factors does not affect bacterial growth [91]. To evaluate the atenolol anti-virulence, it must be tested at sub-MIC to avoid any effect on bacterial growth. For assurance, the effect of atenolol at sub-MIC on the growth of tested strains was evaluated. There was no significant effect of atenolol at sub-MIC on bacterial growth. For attesting the atenolol anti-QS activity, the expressions of *P. aeruginosa* LasI/R, RhlI/R, and PqsA/R QS encoding genes were evaluated in the presence of atenolol at sub-MIC. Interestingly, atenolol significantly downregulated the expression of QS-encoding genes. These findings beside docking results indicate anti-QS activities of atenolol. 

QS controls the regulation of several virulence factors such as motility, biofilm formation, and production of enzymes as protease, hemolysins, urease, elastase, and other virulence factors [5]. CsrA (carbon storage regulator) homologue RsmA is important component of the global regulatory Csr system in *Escherichia coli* that represses a variety of stationary-phase genes [71]. RsmA showed high sequence similarity to *E coli* CsrA and RsmA of *S. marcescens, Erwinia carotovora*, *Haemophilus influenzae* and *Bacillus subtilis* [71,92]. Importantly, RsmA regulates gene expression by affecting the stability of mRNA, and inhibited virulence-gene expression in a complex regulatory network [71,92]. It was shown that transformation of RsmA encoding plasmid to *P. mirabilis* diminished the swarming motility and decreased the production of protease, hemolysins and urease [71]. Interestingly, atenolol significantly upregulated the *rsmA* expression in *S. marcescens*. That is in agreement with the in vitro findings which showed the atenolol significant reduction effect on the production of protease and hemolysins in the three tested bacterial strains. While atenolol significantly increased the *rsmA* expression, it decreased the expression of *rssB* encoding two-component system RssAB that is essential for swarming motility regulation *S. marcescens* [92,93]. The flagellar master regulator operon *flhDC* controls the expression of flagellar proteins [49]. Atenolol downregulated the flagellar transcriptional regulators encoding genes *flhC* and *flhD* in *S. marcescens*. That is in compliance with the current findings that atenolol at sub-MIC significantly crippled the swarming motility of *S. marcescens*, *P. aeruginosa*, and *P. mirabilis*. 

The biofilm formation is an additional obstacle against the efficient antibiotic treatment, and is associated with resistance increment [20]. Atenolol significantly diminished the adhesion biofilm formation in *S. marcescens, P. aeruginosa*, and *P. mirabilis*. That may be interpreted by atenolol downregulation to the fimbria encoding genes *fimA* and *fimC* in *S. marcescens*. The transcriptional factors as BsmA and BsmB are required to increase the expression of type I pilus [74,94]. Atenolol significantly reduced the expression of *bsmB* gene *in S. marcescens*. Finally, the in vivo anti-virulence activity of atenolol was evaluated against *S. marcescens, P. aeruginosa*, and *P. mirabilis*. Atenolol showed significant ability to protect mice from killing. To sum up all the above results, atenolol acquires potent anti-QS and anti-virulence activity and is good candidate to be used in addition of antibiotics in treatment of aggressive infections. 

## 4. Materials and Methods

### 4.1. In Silico Study

#### 4.1.1. Ligand Preparation 

Canonical SMILES of atenolol was retrieved from PubChem database (https://pubchem.ncbi.nlm.nih.gov/) (accessed on 24 May 2022). The 3D structure of the compound was built from the 2D structure of the compound, and energy minimized using EHT forcefield at 0.1 Kcal/mol/Å^2^ gradient RMS on Molecular Operating Environment (MOE 2019.012, Chemical Computing Group ULC, Montreal, QC, Canada). Atenolol 3D Structure was protonated using Protonate 3D module on MOE at physiological pH 7.4. Molecular properties of atenolol were obtained using SWISSADME tool (https://www.expasy.org/resources/swissadme (accessed on 24 May 2022)).

#### 4.1.2. Protein Preparation

In silico molecular docking and visualization were performed for atenolol with the bacterial QS proteins using Molecular Operating Environment (MOE) 2019.0102. The crystal structures of the target proteins (PDB ID:, 3SZT and 6Y4F for *P. aeruginosa* quorum-sensing control repressor, and *P. mirabilis* adhesin MrpH, respectively) were downloaded from the RCSB Protein Data Bank (https://www.rcsb.org/ (accessed on 4 May 2022)). *S. marcescens* QS transcriptional regulator (SmaR) (Uniprot Entry: Q14RS3) has no resolved crystal structure, therefore a SWISS MODEL (https://www.expasy.org/resources/swiss-model, accessed on 28 May 2022) was utilised and active site architecture analyzed for its validation.

The QuickPrep panel was used to prepare the protein structures for the docking process. Preparation involves energy minimization, protonation, fixing and tethering atoms, deleting unnecessary water molecules and initial refinement at gradient RMS of 0.1 Kcal/mol/Å^2^.

Docking for the ligands in the active site was performed using Alpha triangle placement with Amber10: EHT forcefield, refinement with forcefield and scored using the Affinity dG scoring system. Re-docking of the co-crystallized ligand was performed to validate the use of the protein in structure-based drug design.

The binding pocket of each target was defined by the geometrical approach of MOE Site Finder, along with the position of the co-crystallized ligand within the crystal structure of protein. The Computed Atlas for Surface Topography of Proteins (CASTp; http://sts.bioe.uic.edu/castp/index.html, accessed on 28 May 2022) was used to calculate the pocket area/volume across the five QSs proteins [95].

### 4.2. Bacterial Strains, and Chemicals

The clinical isolates *Proteus mirabilis* [49] and *Serratia marcescens* [74] were fully identified previously, in addition to *Pseudomonas aeruginosa* PAO1 (ATCC BAA-47-B1) were used in this study. *Proteus mirabilis* isolate was isolated from macerated diabetic foot infection, and it was recognized as strong biofilm forming [49,67,96]. *Serratia marcescens* was a clinical isolate obtained from endotracheal aspiration [64,97], and was considered a strong biofilm forming [64,98]. *Pseudomonas aeruginosa* PAO1 is known as a strong biofilm forming and used in particular to assess the antibiofilm and anti-QS activities [99,100]. The media were ordered from Oxoid (Hampshire, UK), and the chemicals used were of pharmaceutical grade. Atenolol (CAS numbers: 29122-68-7) were obtained from Sigma-Aldrich (St. Louis, MO, USA).

### 4.3. Effect on Bacterial Growth

The atenolol MICs against the tested strains were detected using the broth microdilution method according to the Clinical Laboratory and Standards Institute Guidelines (CLSI, 2020). 

The atenolol effect at sub-MIC (1/5 MIC) on bacterial growth was evaluated, as formerly described [43,44]. Briefly, the viable counts and optical densities of bacterial cultures provided or not with atenolol at 1/5 MIC were detected.

### 4.4. Adhesion and Biofilm Formation 

The tested strains *S. marcescens* [97,98], *P. aeruginosa* [40,44], and *P. mirabilis* [49,96], were recognized as strong biofilm forming and employed in this study to evaluate the atenolol antibiofilm activity. The crystal violet method was used to assess the antibiofilm activity and adhesion photometrically at 600 nm [44,101]. Briefly, the suspension of tested strains was prepared from overnight cultures and the optical densities was adjusted to OD600 of 0.4 (1 × 10^8^ CFU/mL). Ten μL aliquots of the prepared suspensions were added to 1 mL amounts of fresh tryptone soya broth (TSB) provided or not with atenolol at sub-MIC. Aliquots of 100 μL were transferred into the wells of 96 wells microtiter plates. After 1 h for adhesion assay or overnight incubation for biofilm assay, the planktonic non-adhered cells were aspirated, and the attached cells were fixed with methanol for 20 min and stained with crystal violet (1%) for 30 min. After washing out the excess dye, the attached dye was eluted by glacial acetic acid (33%), and the absorbances were measured at 590.

The atenolol antibiofilm activity was visualized by growing the bacterial strains on cover slips to form biofilms in the presence or absence of atenolol, as formerly detailed [36]. Briefly, the biofilms of different strains in the presence or absence of atenolol at sub-MIC were allowed to be formed on cover slips on 24 well polystyrene plates. After overnight incubation, the excess media and non-adherent cells were washed out and the cover slips were stained with crystal violet and examined.

### 4.5. Bacterial Motility

The bacterial swarming motility was evaluated in the presence of atenolol at sub-MIC in comparison to untreated controls as described previously [49,74]. Briefly, 5 µL of tested bacterial strains were centrally spotted on agar plates provided or not with atenolol at sub-MIC. The swarming zones were measured in mm. 

### 4.6. Protease Production 

The effect of atenolol on the production of protease was evaluated by using the skim milk agar method as formerly described [74]. Briefly, the supernatants were collected from bacterial cultures provided or not with atenolol at sub-MIC. The supernatants containing protease were added to the pre-made wells in 5% skim milk agar plates and incubated overnight at 37 °C, and the clear zones diameters were measured in mm.

### 4.7. Hemolysins Production 

The atenolol anti-hemolytic activity was quantified as earlier described [44]. Briefly, the supernatants were collected from bacterial cultures grown in the presence or absence of atenolol at sub-MIC. Fresh blood (2%) suspensions were mixed in equal volume to the supernatants for 2 h at 37 °C. After the centrifugation, the absorbances were measured at 540 nm. Negative un-hemolyzed blood control and positive completely hemolyzed control were used. 

### 4.8. RT-PCR

The RNA of treated or untreated *S. marcescens* or *P. aeruginosa* with atenolol at sub-MIC was extracted as described before [42]. The primers used to amplify the tested genes were listed in Table 4 indicated previously. The extracted RNA was used to synthesis cDNA, and the relative expression of the tested genes was calculated by the comparative threshold cycle (^ΔΔ^Ct) method [102].

### 4.9. Mice Survival Assay 

The mice survival model was employed to assess the in vivo anti-virulence activity of atenolol as formerly mentioned [42,105]. Briefly, overnight cultures in LB broth provided or not with atenolol at sub-MIC (1 × 10^6^ CFU/mL) in PBS. Four mice groups, each was composed of 10 three-weeks old female Mus musculus mice. Two negative control groups were kept uninfected or intraperitoneally (ip) injected with PBS. The third group was ip injected with untreated bacterial strains as positive control group. The last group was ip injected with atenolol treated bacterial strains. The mice survival was recorded over 5 days using Kaplan–Meier method.

### 4.10. Statistical Analysis

The tests were done in triplicate, and the data are presented as the means ± standard error. The student’s *t*-test was employed to assure the statistical significance (unless mentioned), where *p* value < 0.05 is considered significant (GraphPad Prism Software, v.8, San Diego, CA, USA).

## 5. Conclusions

In the continuous increment of bacterial resistance to antibiotics, alternative strategies could be helpful to conquer the resistance development. Among the most promising approaches, targeting bacterial virulence confers the bacterial attenuation to be easily eradicated by host immunity. This approach does not affect the bacterial growth and hence did not stress bacteria to develop resistance. Due to the known roles of QS in regulation of virulence, anti-QS agents could guarantee significant diminishing of bacterial virulence. Wide diverse chemical moieties were screened for their anti-QS activities; however, the repurposing of the drugs is an advantageous strategy. In the current study, the β-adrenoreceptor antagonist atenolol anti-QS and anti-virulence activities have been studied in silico, in vitro, and in vivo against three Gram-negative bacteria namely *S. marcescens*, *P. aeruginosa*, and *P. mirabilis*. Atenolol showed significant anti-QS, anti-biofilm activities, diminished the production of virulence factors and downregulated the QS and virulence encoding genes. The present findings document the potent anti-QS activities of atenolol and its possible use in addition as adjuvant to traditional antibiotics in treatment of aggressive virulent infections. Taking in consideration that further pharmacological and toxicological studies to ensure their safe use for new application.

## Figures and Tables

**Figure 1 ijms-23-13088-f001:** Atenolol shares metoprolol chemical moiety.

**Figure 2 ijms-23-13088-f002:** (**A**–**C**): Blue-Ribbon representation of the three bio-targets, Putative pockets; white color, SA and V were calculated via the online Computed Atlas of Surface Topography of proteins (CASTp; http://sts.bioe.uic.edu/castp/index.html (accessed on 28 May 2022)), (**D**–**F**): 3D view of the active site indicating the main amino acid residues lining the pocket.

**Figure 3 ijms-23-13088-f003:** (**A**) 3D alignment of Atenolol (Purple) and 3OC12-HSL (Green) in the binding pocket of QScR, (**B**) Binding interactions of Atenolol (Purple) with key residues of QScR (Green). (**C**); 2D ligand interactions of atenolol with key residues of QScR.

**Figure 4 ijms-23-13088-f004:** 3D binding of Atenolol (Purple) with MrpH active site showing the key amino acid interactions along with the crucial Zn^+2^ binding (Turquoise).

**Figure 5 ijms-23-13088-f005:** 3D alignment of Atenolol (Purple) and Glutamate (Green) in the binding pocket of MrpH showing the significant size difference.

**Figure 6 ijms-23-13088-f006:** (**A**); 3D model of *S. marcescens* SmaR, HLC is in thick yellow balls indicating the binding pocket, (**B**); 3D representation of HLC (yellow sticks) in the active site showing interactions with key amino acids (green), H-bond is presented as green dots.

**Figure 7 ijms-23-13088-f007:** (**A**) 3D Atenolol-*S. marcescens* SmaR model interaction diagram, Atenolol is in purple thick sticks within the molecular surface of the active site, amino acid residues of the active site are shown as green thin sticks. H-bond is presented as green dots. *Pi*-bond is presented as turquoise dots, (**B**) 2D ligand interactions between atenolol and the key amino acid residues.

**Figure 8 ijms-23-13088-f008:** Effect of atenolol at sub-MIC on the bacterial growth. (**A**) The optical densities of the bacterial cultures after overnight growth in the presence or absence of atenolol at sub-MIC (**B**) The bacterial viable counts in the presence and absence of atenolol at sub-MIC. There was no significant difference between the viable counts and optical densities of untreated controls and atenolol-treated cultures. ns: nonsignificant (*p* > 0.5).

**Figure 9 ijms-23-13088-f009:** Atenolol decreased the bacterial adhesion and biofilm formation. (**A**) Light microscope images show the marked ability of atenolol to decrease the numbers of adhered bacterial cells to cover slips. (**B**) The absorbances of crystal violet-stained adhered *S. marcescens*, *P. aeruginosa*, and *P. mirabilis*. (**C**) The absorbances of crystal violet-stained biofilm forming *S. marcescens*, *P. aeruginosa*, and *P. mirabilis*. Atenolol significantly decreased the adhesion and biofilm formation (***: *p* ≤ 0.001). The data are presented as percentage change from untreated controls.

**Figure 10 ijms-23-13088-f010:** Atenolol crippled *S. marcescens*, *P. aeruginosa*, and *P. mirabilis*. The effects of atenolol on swarming motilities of tested strains were evaluated. Significantly, atenolol diminished the motility of tested bacterial strains (***: *p* ≤ 0.001). The data are presented as percentage change from untreated controls.

**Figure 11 ijms-23-13088-f011:** Atenolol decreased the production of (**A**) Hemolysins, and (**B**) Protease production in *S. marcescens*, *P. aeruginosa*, and *P. mirabilis*. Atenolol at sub-MIC significantly reduced the production of hemolysins and protease (***: *p* ≤ 0.001). The data are presented as percentage change from untreated controls.

**Figure 12 ijms-23-13088-f012:** Atenolol reduced the expression of virulence and QS genes. The expression of *S. marcescens* motility and adhesion, and *P. aeruginosa* QS genes were evaluated in the presence of atenolol at sub–MIC. The expression of the tested genes was normalized to the expression of housekeeping genes *rplU* and *rpoD* in *S. marcescens* and *P. aeruginosa*, respectively (***: *p* ≤ 0.001).

**Figure 13 ijms-23-13088-f013:** Atenolol protected mice from (**A**) *S. marcescens*, (**B**) *P. aeruginosa*, and (**C**) *P. mirabilis*. Atenolol at sub-MIC showed a significant capacity to protect mice from *S. marcescens*, *P. aeruginosa*, and *P. mirabilis* pathogenesis (log rank test for trend *p* = 0.0289, 0.0025, and 0.0111, respectively). (* *p* < 0.05, *** p* < 0.01).

**Table 1 ijms-23-13088-t001:** Docking results for both 3OC12-HSL and atenolol with *P. aeruginosa* quorum-sensing control repressor.

Ligand	Rigid Receptor Protocol	Induced-Fit Protocol	H-Bond Interactions	Hydrophobic Interactions	Ionic-Bond Interactions
S Score Kcal/mol	RMSD	S Score Kcal/mol	RMSD
Atenolol	−7.4614	1.4216	−7.5851	1.0032	Tyr58 and Gly40	Ser38, Phe54, Tyr52, Tyr58, Tyr66, Val78 and Met127	Asp75
3OC12-HSL	−10.4256	1.2495	−10.1774	1.1676	Tyr58, Trp62 and Asp75	Ser38, Tyr52, Phe54, Tyr58, Tyr66, Val78 and Met127	-

**Table 2 ijms-23-13088-t002:** Docking results for both glutamate and atenolol with *P. mirabilis* adhesin MrpH.

Ligand	Rigid Receptor Protocol	Induced-Fit Protocol	H-Bond Interactions	Hydrophobic Interactions	*pi*-Interactions
S Score Kcal/mol	RMSD	S Score Kcal/mol	RMSD
Atenolol	−6.3937	1.5832	−7.2011	1.1977	Ala84 and Arg118.In addition to ionic bond with zinc metal	Asn82, Ala84, Phe85, Thr115 Thr116, Arg118 and Ile140	Asn82(*pi*-H)
Glutamate	−8.2383	0.9710	−8.4332	1.4964	Asn 82, Thr116, His117 and Arg118.In addition to ionic bond with zinc metal	Asn 82, Thr116, Arg118 and Ile140	-

**Table 3 ijms-23-13088-t003:** Docking results for both HLC and atenolol with *S. marcescens* SmaR.

Ligand	Rigid Receptor Protocol	Induced-Fit Protocol	Interactions
S Score Kcal/mol	RMSD	S Score Kcal/mol	RMSD	H-Bond	Hydrophobic	Ionic	*Pi*
Atenolol	−5.9927	1.2627	−5.7653	1.0260	Asp66	Phe44, Phe54, Tyr57, Asp66, Val68, Trp81 and Ile105	Asp66	Phe44,Tyr57 and Trp81
HLC	−6.9484	1.3639	−7.1761	1.3923	Trp53	Ala32, Phe44, Tyr57, Trp81, Ile105, Val122 and Ser124	-	-

**Table 4 ijms-23-13088-t004:** The primers used in this study.

Target Gene	Sequence (5′–3′)	Gene Significance	Reference
*lasI*	For: CTACAGCCTGCAGAACGACARev: ATCTGGGTCTTGGCATTGAG	*P. aeruginosa* QS autoinducer synthetase	[103]
*lasR*	For: ACGCTCAAGTGGAAAATTGGRev: GTAGATGGACGGTTCCCAGA	*P. aeruginosa* QS receptor	[103]
*rhlI*	For: CTCTCTGAATCGCTGGAAGGRev: GACGTCCTTGAGCAGGTAGG	*P. aeruginosa* QS autoinducer synthetase	[103]
*rhlR*	For: AGGAATGACGGAGGCTTTTTRev: CCCGTAGTTCTGCATCTGGT	*P. aeruginosa* QS receptor	[103]
*pqsA*	For: TTCTGTTCCGCCTCGATTTCRev: AGTCGTTCAACGCCAGCAC	*P. aeruginosa* QS autoinducer synthetase	[104]
*pqsR*	For: AACCTGGAAATCGACCTGTGRev: TGAAATCGTCGAGCAGTACG	*P. aeruginosa* QS receptor	[103]
*rpoD*	For: GGGCGAAGAAGGAAATGGTCRev: CAGGTGGCGTAGGTGGAGAAC	Housekeeping for *P. aeruginosa* genes	[104]
*fimA*	For: ACTACACCCTGCGTTTCGAC Rev: GCGTTAGAGTTTGCCTGACC	*S. marcescens* fimbria	[77]
*fimC*	For: AAGATCGCACCGTACAAACC Rev: TTTGCACCGCATAGTTCAAG	*S. marcescens* fimbria	[77]
*flhc*	For: AAGAAGCCAAGGACATTCAG Rev: TTCCCAGGTCATAAACCAGT	*S. marcescens* flagella	[29]
*flhD*	For: TGTCGGGATGGGGAATATGG Rev: CGATAGCTCTTGCAGTAAATGG	*S. marcescens* flagella	[29]
*bsmB*	For:CCGCCTGCAAGAAAGAACTT Rev: AGAGATCGACGGTCAGTTCC	*S. marcescens* type I pilus	[29]
*rsmA*	For: TTGGTGAAACCCTCATGATT Rev: GCTTCGGAATCAGTAAGTCG	*S. marcescens* motility	[29]
*rssB*	For:TAACGAACTGCTGATGCTGT Rev: GATCTTGCGCCGTAAATTAT	*S. marcescens* motility	[29]
*rplU*	For: GCTTGGAAAAGCTGGACATC Rev: TACGGTGGTGTTTACGACGA	Housekeeping for *S. marcescens* genes	[29]

## Data Availability

Data is contained within the article.

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
