# Peer review of "Characterization of the Anti-Biofilm and Anti-Quorum Sensing Activities of the β-Adrenoreceptor Antagonist Atenolol against Gram-Negative Bacterial Pathogens"

_ijms, 2022, doi:10.3390/ijms232113088_

Round 1
Reviewer 1 Report
Anti-biofilm and anti-virulence activities of atenolol against bacteria Serratia marcescens, Pseudomonas aeruginosa and Proteus mirabilis were evaluated with an in-silico study, and in-vitro and in-vivo studies. It was found that atenolol reduced biofilm formation in vitro. The death rates were also decreased significantly as presented in an in vivo study with mice. The logic of design is clear and the findings contribute to the current field of anti-QS and anti-biofilm activities and evaluation. Moderate revision is needed in regarding to the structure of introduction and English language.
1. Lack of connection between the introduction of QS and bacterial resistance in the first two paragraphs. Transition sentences are needed in between paragraphs to connect the main points.
2. Insufficient information on atenolol as the only testing agent. Although metoprolol and atenolol are both beta-blockers and treat similar diseases. It is confusion if the connection between them is not properly introduced.
3. No details of the specific strain of S. marcescens, P.aeruginosa, and P. mirabilis. This information should be clearly described in Materials&Method. Also no detailed description on the testing method, for example how the biofilm adhesion and formation were evaluated.
4. Typos and errors in Conclusion section. For example, the repurposing of the drug is "an" advantageous strategy; the present findings document the potent "of" atenolol anti-QS activities and its possible use "besides" as an antibiotic "used to treat virulent bacterial infections".
5. Include more discussion on "drug repurposing" in the Discussion section.
Author Response
Dear Reviewer,
We are very thankful for your interest in our manuscript. Please find the attached response to all the raised points.
Best Regards,
Wael

Reviewer 2 Report
Manuscript ID: ijms-1977749 – Review
General comments
This paper by Simona Cavalu, Samar S. Elbaramawi, Ahmed G. Eissa, Mohamed F.
Radwan, Tarek S. Ibrahim, El-Sayed Khafagy, Bruno Silvester Lopes, Mohamed A.
M. Ali, Wael A. H. Hegazy, Mahmoud A. Elfaky focuses on the evaluation of the anti-QS, anti-biofilm and anti-virulence activities of the β-adrenoreceptor blocker atenolol against Gram-negative bacteria. The overall aim of their work is to highlight the anti-QS activities and its possible use beside antibiotics in virulent infections.
The outcome of this study is to give a comprehensive overview of the anti-Biofilm and anti-Quorum sensing activities of the β-Adrenoreceptor antagonist atenolol against Gram-Negative bacterial pathogens. The paper is interesting and fits the scope of ijms. However, there are some points which should be addressed to improve the clarity and quality of the manuscript. An improved revised manuscript would address the following minor comments / suggestions.
Minor Comments
- Introduction
The authors initially describe about quorum sensing in the first paragraph of the introduction section. Although they cite a big number of previous studies, they give only brief and superficial information about the relevance of their study with quorum sensing and how this study relates to the previous ones. I think that they need to elaborate more on quorum sensing of bacteria as this is a big part of their study. Perhaps they can focus on specific papers they cite and state why they refer to them, rather than cite too many altogether within one sentence. The same on the second paragraph regarding to the resistance of bacteria. Generally, the reviewer recommends an extension of the Introduction so that it clearly reflects their study.
- Figure 7
The authors need to describe the figures only and remove from the figure 7 title the result of their statistical analysis and instead, they need to add it to the main text of their paper.
- Figures 8 & 9
The reviewer wants to see a scale bar in the images illustrated in Figures 8 and 9.
- Conclusions
I think that Conclusions need to be re-written from the Authors. If a reader focuses on conclusions, it will be impossible to understand the importance of this study. The authors need to consider how important is this section of the paper and try make a greater effort to write to the point conclusions of their work.
Author Response

(The authors gave the same response as above.)

Reviewer 3 Report
The manuscript entitled :
Characterization of the Anti-Biofilm and Anti-Quorum sensing activities of the β-Adrenoreceptor Antagonist Atenolol against Gram-Negative Bacterial pathogens
By Simona Cavalu et al reports studies designed to evaluate the anti-QS, anti-biofilm and anti-virulence activities of the β-adrenoreceptor blocker atenolol against Gram-negative bacteria Serratia marcescens, Pseudomonas aeruginosa, and Proteus mirabilis. After an in-silico study to evaluate the binding affinity of atenolol to S. marcescens SmaR QS receptor, P. aeruginosa QscR QS receptor, and P. mirabilis MrpH adhesin the anti-virulence activity was evaluated against these strains in-vitro and in-vivo.
This work aims at repurposing drugs to develop the use of them as antivirulence compounds. This is an interesting topic but this work raises some concerns:
In figures 8, 9, and 10, even if three different bacterial species are tested, the authors present their results with only one control presented as “Untreated bacteria”: this set of data has no error bar. This referee wonders how “p” values have been calculated as each treated bacteria should have its own control data set with its own distribution and error.
The second major concern is about the in vivo test the authors in their discussion wrote “Atenolol showed significant ability to protect mice from killing”
However only the bacteria were treated with Atenol and not the mice themselves prior infection. So, the author should discuss more about the possibility to use Atenol as a drug i.e. to be delivered to mice to protect them against virulent bacteria. What is the toxicity of Atenol and is it compatible with the assays presented here?
Minor points:
Line 98: Qs control repressor: give here the name of the protein
Line 111 define LBD and DBD
Line 128 define HLC
The paragraph from line 121 to 130 is unclear and need rewriting
Figure 3: the inset is of very bad quality and it is not possible to read it
Legend of figure 8: replace B by C
Figure 11: indicate the meaning of red and green
Author Response

(The authors gave the same response as above.)

Round 2
Reviewer 1 Report
The introduction was improved.
Reviewer 3 Report
No additional comments, thanks for answering the questions